# Differential Magnitude of Resilience between Emotional Intelligence and Life Satisfaction in Mountain Sports Athletes

**DOI:** 10.3390/ijerph20156525

**Published:** 2023-08-04

**Authors:** Guillermo Sanz-Junoy, Óscar Gavín-Chocano, José L. Ubago-Jiménez, David Molero

**Affiliations:** 1Spanish Federation of Mountain Sports and Climbing, 08015 Barcelona, Spain; sistemagestion@fedme.es; 2Department of Pedagogy, Lagunillas Campus, University of Jaén, 23071 Jaén, Spain; ogavin@ujaen.es; 3Department of Didactics of Musical, Plastic and Corporal Expression, University of Granada, 18071 Granada, Spain; jlubago@ugr.es

**Keywords:** emotional intelligence, resilience, life satisfaction, mountain sport

## Abstract

The field of mountain sports has its own characteristics, different from other sports modalities. Emotional intelligence and resilience not only refer to the adaptive capacity that can be developed after an adverse experience, but can also be factors that positively affect sporting performance in extreme conditions. In this study, 4818 athletes from the Spanish Federation of Mountain Sports and Climbing participated; 2696 were men (67.1%) and 1322 were women (32.9%), with a mean age of 49.42 years (±11.9). The Resilience Scale (RS-14), Wong Law Emotional Intelligence Scale (WLEIS-S) and Satisfaction with Life Scale (SWLS) were used. The aim was to provide evidence for the potential for resilience (personal competence and acceptance of self and life) among EI and life satisfaction in mountain and climbing athletes. The results showed that the coefficients of determination of personal competence [(*Q*^2^ = 0.286); (*R*^2^ = 0.578)], acceptance of self [(*Q*^2^ = 0.310); (*R*^2^ = 0.554)] and life satisfaction [(*Q*^2^ = 0.299); (*R*^2^ = 0.421)] for the estimation of the measurement model indicated a good model fit. In the future, it would be necessary to carry out specific studies discriminating by sport modality and methods for this area, with a large number of athletes and disciplines, as well as evaluating its possible applications for the improvement of emotional factors.

## 1. Introduction

Mountain sports have experienced significant growth worldwide, especially in the last decade. In this regard, a recent study of sports habits in Spain [1] reports that 15.8% of the population practised hiking and/or mountaineering during 2020 (more than 7,400,000 people), a significant increase on the data reported by Pérez and Luque-Valle [2] before the start of the COVID-19 pandemic.

Mountain sports have a sporting environment with its own characteristics, different from other sports modalities and risk conditions in some cases, which may affect the profile of athletes physically, but above all psychologically. The pandemic has produced some changes in behaviour and in the ways of participating in sport in the Spanish population [3]; hiking practitioners have grown exponentially, but not all of them are federated [4]. In this regard, we present this study, which takes as participants those people who practise mountain sports and who are linked to the only Spanish sports federation that will participate in the Summer Olympic Games (climbing) and in the Winter Olympic Games through ski mountaineering, which is debuting as an Olympic discipline at Milan–Cortina d’Ampezzo 2026. There are still few studies analysing resilience, emotional intelligence (EI) and its relationship with life satisfaction in mountain sports, due to the characteristics of the discipline. EI and resilience not only refer to the adaptive capacity that can be developed after an adverse experience, but can also positively affect sporting performance in extreme conditions [5,6].

### 1.1. Resilience

Resilience is defined as the ability to exhibit adaptive responses to adverse situations [7]. According to Wagnild and Young [8], resilience has two factors and it is appropriate to clearly define them. The first is personal competence. This is understood as knowledge of factors of personal capacity, self-confidence, independence, decisiveness, invincibility, power, ingenuity, skill and perseverance. This factor basically seeks to assess self-confidence. The second is self-esteem and self-worth, which allow a feeling of peace to be experienced in spite of problems [9]. Resilience is a factor related to emotions, and generates determination, self-control and the ability to solve problems successfully. The relationship between EI and resilience suggests the presence of greater well-being to cope with experiences of adversity and develop personal competence [5,10].

Although this construct has been extensively researched in different fields, such as in clinical and general psychology; there has not been as much interest from professionals specialising in sport performance, which is striking because adversity and stress (in acute and chronic forms) are common in this context. Resilience in mountain athletes has been analysed in some studies [11,12,13,14], and its connection with behavioural addiction to extreme mountain sports has been associated with high levels of resilience and emotional regulation management, as well as increased risk-taking [6,15].

### 1.2. Emotional Intelligence

The concept of EI was first introduced in an article published by Salovey and Mayer [16]. However, the conceptualisation of the EI construct is a matter that requires consensus among researchers. Petrides and Furnham [17] distinguish two different EI constructs: EI as a personality trait, on the one hand, and EI as an ability on the other. EI as an ability has been conceived as “an ability to perceive, assimilate, understand and regulate one’s own and others’ emotions, promoting emotional and intellectual growth” [18] (p. 10). EI as a capacity should be measured through tests of peak performance, whereas EI as a trait would refer to a constellation of behavioural dispositions and self-perceptions concerning one’s abilities to recognise, process and use emotionally charged information [19]. Among studies on mountaineering and outdoor sports athletes, we highlighted those that analysed the use of emotional regulation strategies [20], EI and emotional competences of athletes [21,22,23], EI in mountain trainers [5] and the influence of EI on climbing performance [24,25].

### 1.3. Well-Being and Life Satisfaction

One of the most productive fields of EI research focuses primarily on providing evidence of the relationship with psychological well-being and life satisfaction [26], which refers to the state of the individual in which both objective and subjective needs are satisfied [27]. Although there are different psychological approaches to this concept, it is subjective well-being which has perhaps awakened the greatest interest in social psychology over the last few years [28]. As Triadó et al. [29] point out, subjective well-being studies people’s emotional experiences, the satisfaction of different vital domains and the global assessment of one’s own life. Sánchez-Álvarez et al. [30] highlighted how the appropriate use of certain emotional strategies could contribute to experiencing a higher rate of positive emotional states and the reduction in negative emotional states; therefore, this would have a positive impact on people’s well-being and health. In relation to life satisfaction and optimism in mountain sports [31,32], we highlight the work of Próchniak [33] and the one based on personality and emotional responses in mountain hikers [34]. Frochot et al. [35] analysed the satisfaction of athletes in these mountain sports disciplines and the self-perceived well-being that this activity produces in mountain tourism contexts.

There is extensive evidence of the well-being generated by contact with nature for physiological and mental health, especially in well-conserved ecosystems with a high level of biodiversity, such as protected natural areas [36]. The systematic review conducted by Sandifera et al. [37] provided a comprehensive review of the psychological and cognitive benefits, in addition to the physiological, social and disease-regulating benefits of interacting with well-preserved natural areas. The meta-analysis conducted by Kelley et al. [28] reported that physical activity in the natural outdoor environments was associated with higher well-being. Outdoor sporting activity in a natural setting improves self-esteem and is more restorative than in an urban environment. The moods and attitudes of participants in outdoor sports activities are significantly positive after participating in activities compared to indoor activities [36,38]. In this regard, the work of Engemann et al. [39] revealed that the risk of psychological disorders from adolescence to adulthood decreases according to the amount of green space near a person’s residence.

### 1.4. Effect of Emotional Intelligence on Resilience and Life Satisfaction

The positive effects of EI and adaptive response or resilience, related to life satisfaction, may promote effective coping strategies in adverse situations [40,41]. If we combine the analysis of these variables, it has been shown that people with high EI scores are more satisfied with life [40,42,43]. Although the positive influence of EI on life satisfaction has been reported in different studies, the enhancing effect on resilience has not been considered [41,44,45]. Although this relationship has been analysed in different fields, such as psychology, social sciences and in general, there has not been as much interest in mountain and climbing athletes, which is striking, since adversity and stress are common in this context. However, it is necessary to point out that EI is a factor of psychological adjustment associated with well-being [46], as well as a key variable in personal and social growth [47]. It is also key in the adaptive process, and in social and emotional learning (SEL) understood as the process of integrating cognition, emotion and behaviours throughout a lifetime [48].

### 1.5. Study Purpose and Hypotheses

Based on the initial approach, the aim of this study was to provide evidence for the potential effect of resilience (personal competence and self- and life-acceptance) on EI and life satisfaction in mountain and climbing athletes in Spain. The following hypotheses are considered as working hypotheses (See Figure 1). Resilience is a factor related to emotions, which generates determination, self-control and the ability to solve problems in a positive way. The relationship between EI and resilience indicates the presence of physical, psychological and social resources to face experiences of adversity and develop personal competence [10].

**Hypothesis** **1** **(H1)**.
*EI will be positively related to the dimension of resilience (personal competence).*


One of the keys to the relationship between EI and resilience relates to the fact that stressful events are highly emotionally charged. People’s ability to regulate emotions is a key factor in their acceptance of self and life [49].

**Hypothesis** **2** **(H2)**.
*EI will be positively related to the dimension of resilience (acceptance of self and life).*


The positive effects of EI and the adaptive response or resilience, related to life satisfaction, may favour effective coping strategies against adverse situations [40].

**Hypothesis** **3** **(H3)**.
*The resilience dimension (personal competence), as a potential of EI, will be related to life satisfaction.*


Different research works have considered that high scores in EI and resilience, as self- and life-acceptance, are related to psychological well-being and life satisfaction [50,51,52].

**Hypothesis** **4** **(H4)**.
*The dimension of resilience—the acceptance of self and life—as a potential of EI will be related to life satisfaction.*


## 2. Materials and Methods

### 2.1. Participants

Participants were selected from a population of 107,588 people through probabilistic sampling (estimated for an error of 2% and a confidence level of 99%, with a total of 4006 cases). The final sample obtained was higher than the estimated sample, and it was composed of 4818 athletes over 18 years of age (age of majority in Spain), federated in the Spanish Federation of Mountain Sports and Climbing, with a valid federation licence during the year 2022, from the 17 Spanish autonomous communities (regions) and the 2 autonomous cities of Spain. In relation to gender distribution, 2696 subjects were male (67.1%) and 1322 were female (32.9%). The mean age of the participants was 49.42 years old (±11.9).

In relation to the type of discipline they practiced within mountain sports, 31.68% practiced hiking, 15% mountaineering, 11.36% mountain running, 10.03% climbing or para-climbing, 7.59% climbing on climbing walls, 5,32% snowshoeing, 7.25% ski mountaineering, 3.35% canyoning, 2.12% Nordic walking, 1.52% ice climbing and 0.49% snow running, and 4.02% practice other mountain disciplines.

### 2.2. Instruments

The Spanish version of the Resilience Scale (RS-14) of Wagnild [53] was developed by Sánchez-Teruel and Robles-Bello [54]. It measures the degree of resilience, which is considered to be a positive personality characteristic that allows the individual to adapt to adverse situations. The RS-14 measures two dimensions: personal competence (11 items ) and acceptance of self and life (3 items).

The Spanish version of the Wong Law Emotional Intelligence Scale (WLEIS-S) was used to assess EI [55]. It is based on the Wong and Law EI Scale (WLEIS) [56]. It consists of 16 items and 4 dimensions: intrapersonal perception (evaluation of one’s own emotions), interpersonal perception (evaluation of the emotions of others), assimilation (use of emotions) and emotional regulation. A 7-point Likert-type scale (1 to 7 points) was used, with a validity and reliability in Spanish contexts of (α = 0.91).

The Satisfaction With Life Scale (SWLS) [57] was used, which was the version of the Satisfaction With Life Scale developed by Vázquez et al. [58]. It assesses the satisfaction with life scale. It consists of five items, and participants must indicate the degree of agreement or disagreement for each of the instrument’s response options. The scale in the Spanish version reports an internal consistency of α = 0.82.

### 2.3. Procedure

The ethical guidelines promoted and encouraged by national and international regulations for conducting research with people were followed, through the use of informed consent and the guarantee of confidentiality and anonymity of the data obtained. All data were processed in accordance with EU Regulation 2016/679 of the European Parliament and of the Council, 27 April 2016, both on Personal Data, and Spanish Organic Law 3/2018, of 5 December, regarding the guarantee of digital rights. The data were collected and their quality checked, ensuring at all times that the process complied with the ethical principles for research defined in the Declaration of Helsinki [59] and the standards of integrity in research of The European Code of Conduct for Research Integrity [60]. The instrument was administered individually through the Google platform (Google LLC) between October and December 2022. The approximate response time for each subject was 20 min. This research was approved by the Ethics Committee for Research on Human Subjects of the University of Jaén (Spain), ID code OCT.22/2-LINE.

### 2.4. Data Analysis

Descriptive statistics (means and standard deviations) were obtained, with the a priori analysis of the validity, reliability (Cronbach’s alpha and McDonald’s Omega coefficient) and internal consistency of each instrument, through Confirmatory Factor Analysis (CFA), to verify the psychometric properties of the instruments and determine the factor loadings of each item (see Appendix A). The normality analysis was performed using multivariate hypothesis testing to reduce problems in parameter estimates (e.g., standard errors), which resulted in a normal distribution and fulfilled the assumptions for parametric tests (independence, normality, homoscedasticity). The analyses were performed using the SPPS AMOS 25 program and the Jamovi software (The Jamovi Project) Version 1.2 and Smart-PLS (version 3.3.6). In relation to the coefficients considered in this study, the Chi-square test (*χ*^2^), the degrees of freedom (*df*) and the CFI, GFI, SRMR and RMSEA fit indices were used. In this regard, *χ*^2^ should be understood from the ratio in relation to the degrees of freedom (*χ*^2^/*df*), where the values should be between 2 and 5. The Comparative Fit Index (CFI) measures the relative fit of the observed model, whose value should be greater than 0.90 to indicate a good fit. Similarly, a Goodness of Fit Index (GFI) result above 0.90 shows the proportion of variance and covariance of the model data. Similarly, the Standardized Root Mean Square Residual (SRMR), standardized means of the residuals, i.e., the difference between the observed and model matrix, indicates a good fit of the model when it is below 0.10. The Root Mean Square Error of Approximation (RMSEA) per degree of freedom, as a measure of discrepancy, should have results below 0.08 [61]. A 95% confidence level (significance *p* < 0.05) was used in all cases.

## 3. Results

### Structural Model

To assess the robustness of the factor loadings and the significance between variables, the Bootstrapping procedure was used with 2000 subsamples [62], which resulted in the structural model (Figure 2), in which the variables considered in this study are reported. The predictive significance and standardised regression coefficient or path coefficient of personal competence [(*Q*^2^ = 0.286); (*R*^2^ = 0.578)], acceptance of self and life [(*Q*^2^ = 0.310); (*R*^2^ = 0.554)] and life satisfaction [(*Q*^2^ = 0.299); (*R*^2^ = 0.421)], in the estimation of the measurement model, indicated a good model fit. In this regard, *R*^2^ values above 0.67 indicated a substantial model fit and above 0.33 a moderate fit [63,64].

Table 1 reports Cronbach’s alpha, Omega coefficient, external loadings and Composite Reliability Index (CFI) scores. In relation to the convergent validity or degree of certainty regarding the proposed indicators measuring the same latent variable or factor, through the estimation of the Average Variance Extracted (AVE). The values must be greater than 0.5, according to the criteria of Becker et al. [65]. That is, a high value of (AVE) will have a better representation of the loading of the observable variable (see Table 1).

Discriminant validity (Table 2) measures the difference between the latent variable in order to determine the statistical differentiation of each factor with respect to the others, with the square root of the mean variance extracted in bold [66].

Discriminant validity (Table 3) was examined by analysing the cross-loadings of each of the latent variables and their respective observed variables, where the loadings were higher than the rest of the variables [67].

Table 4 shows the results of the hypothesis testing, following the criteria of Hair et al. [62], where the causal relationship with the latent variables can be observed. The *t*-test obtained values higher than 1.96, which indicated the coherence of the model. In this research, the results that showed a higher value were acceptance of self and life -> life satisfaction (*β* = 0.230, *t* = 13.065, *p* < 0.001); appraisal of one’s own emotions -> acceptance of self and life habits (*β* = 0.149, *t* = 9.489, *p* < 0.001); appraisal of one’s own emotions -> personal competence (*β* = 0.161, *t* = 7.152, *p* < 0.001); appraisal of others’ emotions -> personal competence (*β* = 0.149, *t* = 10.784, *p* < 0.001); personal competence -> acceptance of self and life (*β* = 0.430, *t* = 24.970, *p* < 0.001); personal competence -> life satisfaction (*β* = 0.466, *t* = 29.078, *p* < 0.001); regulation of emotion -> personal competence (*β* = 0.144, *t* = 4.155, *p* < 0.001); use of emotion -> acceptance of self and life (*β* = 0.268, *t* = 15.520, *p* < 0.001); and use of emotion -> personal competence (*β* = 0.455, *t* = 23.397, *p* < 0.001).

## 4. Discussion

The purpose of this research was to provide empirical evidence for the relationship between EI, resilience and life satisfaction in mountain and climbing athletes. Firstly, the results obtained show that EI showed a significant relationship with life satisfaction, where resilience acted as a mediating variable. According to the initial hypotheses, consistent with other research and theoretical frameworks [30,40,43,47], EI dimensions were positively related to life satisfaction, which acted as a mediating variable of resilience (acceptance of self and life and acceptance and personal competence). Although different research works related to high-level sport have considered the relationship between EI and resilience as factors to be developed, and not only as protective dimensions in which athletes face challenges and tolerate adversity, leading to greater life satisfaction [68], they are also conditioned by psychosocial aspects [69].

According to the first hypothesis (H1), EI was related to the resilience variable (personal competence), with the use of emotion dimension scoring the highest. Personal competence is a factor that directly affects the emotional area, providing organisation, determination, self-control and the ability to solve problems in a positive way; resilience, in mountain athletes, as a psychosocial process, should combine psychological and social aspects such as management processes and emotional use, as well as interactions with other people that help to provide strength in high-level disciplines [5,70].

In relation to the second hypothesis (H2), the use of the emotion dimension of EI was related to the dimension of resilience (acceptance of oneself and life). Resilience develops from personal growth in adverse circumstances [71]. In the field of sport, within a framework of reference for the promotion of personal growth in high-level sporting disciplines, intrapersonal resilient resources should be considered as they are beneficial in sporting activities with a high level of self-improvement and athlete psychological well-being [69,72,73].

Regarding the third hypothesis (H3), the dimension of resilience (personal competence), acting as a mediating variable of EI, is positively related to life satisfaction. Different research supports these results, where resilient athletes who are more satisfied with life predict positively and significantly higher EI [47]. In relation to these aspects, several elements coexist that connect resilience and life satisfaction, such as health, sports performance and context, as well as the emotions experienced in personal activities and relationships [3,21,22,23], so that athletes who show higher personal competences also show higher life satisfaction, where resilience plays a mediating role with EI [47,73].

Finally, the third hypothesis (H4), the resilience dimension (acceptance of self and life), was related to life satisfaction, although it had less impact on the results. That is, the dimensions (appraisal of others’ emotions and regulation of emotion) were not related to the resilience dimension (acceptance of self and life). Different studies established that the dimension (acceptance of self and life) was not significantly related to life satisfaction [74]. Other research established that this relationship may be conditioned by other variables, such as self-concept [5,43].

Among the limitations of this research on mountain and climbing athletes that should be taken into account for future research, we highlight the following. Firstly, self-report instruments for EI, resilience and life satisfaction were used. Another limitation of the research is the lack of differentiation between mountain and climbing sports disciplines, which could affect the results. Consequently, caution should be exercised in the generalisation of these data. It would have been interesting to analyse in detail differences in relation to gender and context.

Despite these limitations, this research makes a necessary contribution to the fields of EI, resilience and their influence on life satisfaction, based on personal competence and acceptance of oneself and life. On the other hand, the practical consequences of this work underline the need to strengthen emotional and resilient strategies in athletes with a high level of demand in order to improve personal well-being.

## 5. Conclusions

To conclude, the results of this study could suggest (a) that the dimensions of EI and resilience in mountain and climbing athletes are unequally related to life satisfaction, with the potential level of resilience as a determining factor between EI and life satisfaction, depending on contextual factors or the nature of the group under study; (b) that emotional use and regulation are not conditioning factors of the dimension (acceptance of oneself and of life), and that there is empirical evidence that considers these circumstances; (c) that personal competences are related to life satisfaction in a determinant way; and (d) that both EI and resilience are conditioning factors for higher life satisfaction. Therefore, it could be pointed out that the results of this research offer evidence to understand the complementarity between the dimensions of EI and resilience as determinants of greater life satisfaction.

The practical implications of this research suggest that improving athletes’ emotional intelligence could have a positive impact on their psychological well-being and life satisfaction. In this sense, specific emotional training programs could be developed for athletes that address skills such as the recognition and regulation of emotions, empathy and assertiveness. Similarly, the importance of resilience as a key ability in facing and overcoming challenges and adversities in the context of mountain and climbing sports should be addressed [75]. Resilience training strategies could be implemented to help athletes develop greater adaptive and recovery capacities [76].

The findings of this research may be relevant in terms of psychological evaluation in the sports field, not only focusing on physical performance, but also on emotional and well-being aspects. Valid and reliable measurement instruments could be used to assess the emotional intelligence, resilience and life satisfaction of athletes, which would make it possible to identify areas for improvement and design personalized interventions, understanding that life satisfaction is a key aspect for the general well-being of athletes. Coaches and sports managers could consider strategies that foster a positive and motivating environment in the sports context, in addition to providing emotional support to athletes to improve their satisfaction with sports practice.

## Figures and Tables

**Figure 1 ijerph-20-06525-f001:**
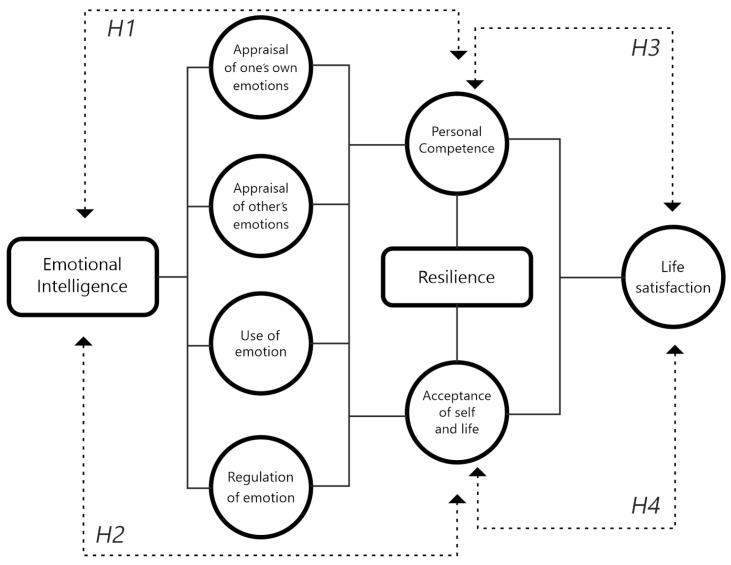
Proposed theoretical model.

**Figure 2 ijerph-20-06525-f002:**
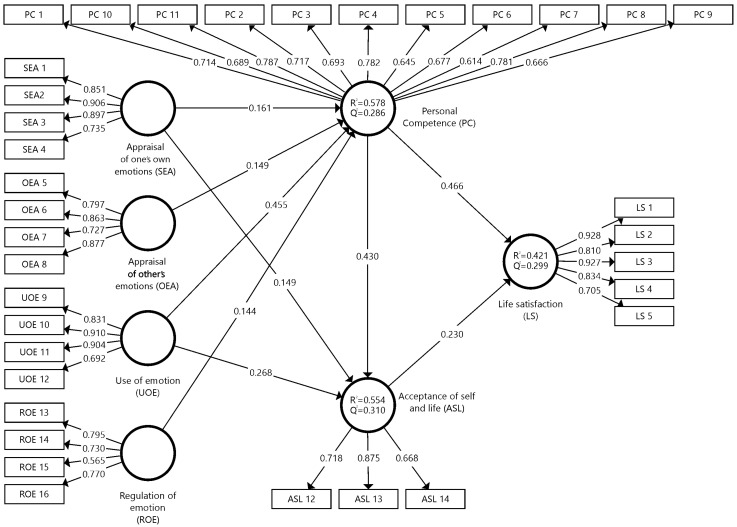
Model reliability and validity.

**Table 1 ijerph-20-06525-t001:** Convergent validity.

Variable	*α*	Composite Reliability Index (CFI)	Rho_A	Average Extracted Variance (AVE)
Acceptance of self and life (ASL)	0.630	0.801	0.686	0.576
Appraisal of one’s own emotions (SEA)	0.869	0.912	0.877	0.723
Appraisal of others’ emotions (OEA)	0.833	0.890	0.846	0.669
Life satisfaction (LS)	0.897	0.925	0.909	0.714
Personal competence (PC)	0.900	0.917	0.905	0.501
Regulation of emotion (ROE)	0.689	0.810	0.715	0.519
Use of emotion (UOE)	0.856	0.904	0.878	0.704

Note: *α =* Cronbach’s alpha.

**Table 2 ijerph-20-06525-t002:** Discriminant validity.

Variable	1	2	3	4	5	6	7
1. Acceptance of self and life (ASL)	**0.759**						
2. Appraisal of one’s own emotions (SEA)	0.533	**0.850**					
3. Appraisal of others’ emotions (OEA)	0.301	0.435	**0.818**				
4. Life satisfaction (LS)	0.558	0.454	0.304	**0.845**			
5. Personal competence (PC)	0.702	0.578	0.447	0.628	**0.708**		
6. Regulation of emotion (ROE)	0.628	0.853	0.593	0.529	0.714	**0.721**	
7. Use of emotion (UOE)	0.641	0.503	0.312	0.520	0.692	0.755	**0.839**

Note: The bold highlights the highest values of each instrument and dimension.

**Table 3 ijerph-20-06525-t003:** Cross-loadings (latent and observable variables).

Variable	Acceptance of Self and Life	Appraisal of One’s Own Emotions	Appraisal of Others’ Emotions	Life Satisfaction	Personal Competence	Regulation of Emotion	Use of Emotion
ASL 12	**0.718**	0.356	0.175	0.345	0.434	0.373	0.328
ASL 13	**0.875**	0.504	0.249	0.543	0.663	0.598	0.614
ASL 14	**0.668**	0.326	0.259	0.344	0.463	0.420	0.472
SEA 1	0.423	0.851	0.365	0.361	0.469	0.655	0.405
SEA 2	0.503	**0.906**	0.381	0.403	0.523	0.795	0.465
SEA 3	0.483	**0.897**	0.388	0.404	0.506	0.718	0.444
SEA 4	0.394	**0.735**	0.342	0.373	0.465	0.729	0.391
OEA 5	0.251	0.353	**0.797**	0.279	0.360	0.484	0.271
OEA 6	0.245	0.355	**0.863**	0.236	0.384	0.565	0.259
OEA 7	0.181	0.299	**0.727**	0.188	0.306	0.364	0.200
OEA8	0.296	0.408	**0.877**	0.283	0.405	0.511	0.285
LS 1	0.531	0.451	0.289	**0.928**	0.576	0.518	0.500
LS 2	0.436	0.353	0.232	**0.810**	0.485	0.403	0.383
LS 3	0.532	0.450	0.291	**0.927**	0.576	0.517	0.500
LS 4	0.457	0.359	0.250	**0.834**	0.560	0.437	0.450
LS 5	0.383	0.281	0.211	**0.705**	0.441	0.338	0.342
PC 1	0.436	0.381	0.330	0.461	**0.714**	0.475	0.438
PC 2	0.492	0.380	0.304	0.531	**0.717**	0.487	0.520
PC 3	0.446	0.354	0.348	0.415	**0.693**	0.475	0.474
PC 4	0.518	0.424	0.329	0.457	**0.782**	0.549	0.556
PC 5	0.413	0.353	0.299	0.352	**0.645**	0.458	0.447
PC 6	0.468	0.410	0.353	0.365	**0.677**	0.495	0.487
PC 7	0.401	0.388	0.298	0.373	**0.614**	0.449	0.382
PC 8	0.670	0.495	0.272	0.512	**0.781**	0.611	0.641
PC 9	0.440	0.361	0.354	0.313	**0.666**	0.457	0.420
PC 10	0.539	0.456	0.294	0.576	**0.689**	0.512	0.460
PC 11	0.574	0.469	0.337	0.471	**0.787**	0.559	0.508
ROE 13	0.502	0.605	0.380	0.403	0.522	**0.795**	0.464
ROE 14	0.395	0.730	0.343	0.373	0.465	**0.735**	0.391
ROE 15	0.245	0.354	0.565	0.236	0.384	**0.862**	0.259
ROE 16	0.593	0.467	0.281	0.470	0.639	**0.909**	0.707
UOE 9	0.391	0.326	0.270	0.353	0.467	0.465	**0.692**
UOE 10	0.546	0.410	0.243	0.440	0.568	0.567	**0.831**
UOE 11	0.593	0.468	0.281	0.471	0.640	0.770	**0.910**
UOE 12	0.594	0.467	0.263	0.469	0.629	0.694	**0.904**

Note: The bold highlights the highest values of each instrument and dimension.

**Table 4 ijerph-20-06525-t004:** Path coefficient (standardized regression coefficient).

Relationship Between Variables	*β*	*SD*	*t*	*p*
Acceptance of self and life (ASL) -> Life satisfaction (LS)	230	0.018	13.065	***
Appraisal of one’s own emotions (SEA) -> Acceptance of self and life (ASL)	0.149	0.016	9.489	***
Appraisal of one’s own emotions (SEA) -> Personal competence (PC)	0.161	0.023	7.152	***
Appraisal of others’ emotions (OEA) -> Personal competence (PC)	0.149	0.014	10.784	***
Personal competence (PC) -> Acceptance of self and life (ASL)	0.430	0.017	24.970	***
Personal competence (PC) -> Life satisfaction (LS)	0.466	0.016	29.078	***
Regulation of emotion (ROE) -> Personal competence (PC)	0.144	0.035	4.155	***
Use of emotion (UOE) -> Acceptance of self and life (ASL)	0.268	0.017	15.520	***
Use of emotion (UOE) -> Personal competence (PC)	0.455	0.019	23.397	***

Note: *β*: Path coefficient; *SD*: standard deviation; *t*: *t* Student; *** = *p* < 0.001.

## Data Availability

The datasets generated during and/or analysed during the current study are available from the corresponding author on reasonable request.

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
