# Peer review of "Differential Magnitude of Resilience between Emotional Intelligence and Life Satisfaction in Mountain Sports Athletes"

_ijerph, 2023, doi:10.3390/ijerph20156525_

Round 1

Reviewer 1 Report

At the outset, I would like to thank you for the opportunity to do a review of your work.

The authors have done well with the task they have undertaken.

The article meets the criteria of science and fits into the theme of the journal.

The quality is at a high level.

I suggest that the authors refer to the following papers in the discussion:

- https://doi.org/10.3389/fpsyg.2023.1120238

Author Response

Reviewer 1 (Review Report, round 2)

Reviewer 1: Black.

Authors: Red.

Reviewer 1:

At the outset, I would like to thank you for the opportunity to do a review of your work.

The authors have done well with the task they have undertaken.

The article meets the criteria of science and fits into the theme of the journal.

The quality is at a high level.

Authors:

Thank you for all the comments and positive evaluations of our study.

Reviewer 1: 

I suggest that the authors refer to the following papers in the discussion:

- https://doi.org/10.3389/fpsyg.2023.1120238

Authors:

The article suggested by the reviewer has been included in the discussion of the article as well as in the references [73].

73.

Piepiora, P.; BagiÅ„ska, J.; Witkowski, K.; Nakonieczna, J; Piepiora, Z. Comparison of personality differences of Polish mountaineers. Frontiers in psychology 202314, 1120238. https://doi.org/10.3389/fpsyg.2023.1120238

Reviewer 2 Report

This manuscript was looking into the on the relationship between emotional intelligence, resilience and life satisfaction in mountain and climbing athletes. They managed to collect quite impressive sample of 4818 athletes. Their findings provide better insight into the dimensions of emotional intelligence and resilience as determinants of higher life satisfaction.

However, I have few suggestions/comments.

At first, I thought your sample consisted of professional athletes. Correct me if I am wrong, but that does not seem to be a case. It is important to make a distinction between professional and recreational/amateur athletes. Interpretation of the findings will dramatically vary depending on which group you are referring to. Along those lines, I noticed that mean age of your sample is quite on the high end (49.42). If we look at most finding from the literature, both emotional intelligence and life satisfaction have tendency to rise with age. I was wondering if you have controlled for age when doing your stats? If not, it would be interesting to see whether age and gender have any impact on your sample findings. 

While I can agree that your manuscript is well written, I believe it would greatly profit from some reformatting. Current version is written like a Master thesis. Which is not bad. But for scientific publication, format is bit different. I would like to see more expanded discussion and that you connect with previous and current literature. Now, you just go over your hypothesis in few sentences and that is it. 

I would put limitation section into discussion and leave Conclusion section purely for your results and implications. Short and concise. Also, mention age in your limitation section. If we are talking about athletes, most of the time we tend not to think of 5o years old. Your results could maybe differ drastically if your mean age was for example 34. 

Minor comments:

Line 108: title of the section is: Effect of emotional intelligence on resilience and satisfaction with life. Please change satisfaction with life to life satisfaction. 

Section 2.1.: typo Parcipants

Table 4.: Missing 1st letter in the first variable.

Author Response

Reviewer 2

Reviewer 2: Black.

Authors: Red.

Reviewer 2:

This manuscript was looking into the on the relationship between emotional intelligence, resilience and life satisfaction in mountain and climbing athletes. They managed to collect quite impressive sample of 4818 athletes. Their findings provide better insight into the dimensions of emotional intelligence and resilience as determinants of higher life satisfaction.

Authors:

We appreciate the reviewer's evaluations, and their satisfaction with our article. We agree with the reviewer that one of the strengths of our research is the sample size.

Reviewer 2:

However, I have few suggestions/comments.

Authors:

We appreciate all your comments and suggestions

Reviewer 2:

At first, I thought your sample consisted of professional athletes. Correct me if I am wrong, but that does not seem to be a case. It is important to make a distinction between professional and recreational/amateur athletes. Interpretation of the findings will dramatically vary depending on which group you are referring to. Along those lines, I noticed that mean age of your sample is quite on the high end (49.42). If we look at most finding from the literature, both emotional intelligence and life satisfaction have tendency to rise with age. I was wondering if you have controlled for age when doing your stats? If not, it would be interesting to see whether age and gender have any impact on your sample findings. 

Authors:

In previous studies, we have carried out analyzes to establish the existence of significant differences based on the gender and age of the participants, as suggested by the reviewer. In this sense, you can consult:

Gavin-Chocano, O.; Martin-Talavera, L.; Sanz-Junoy, G.; Molero, D. Emotional Intelligence and Resilience: Predictors of Life Satisfaction among Mountain Trainers. Sustainability 2023, 15, 4991. https://doi.org/10.3390/su15064991

Age in mountain and climbing sports practitioners is not an explicit condition for their development, as shown by data from the Spanish Federation of Mountain and Climbing Sports. In the description of the participants it is specified that the subjects of the sample are federated athletes in mountain sports and climbing, and it is not essential that they participate in competitive activities. Only two disciplines, ski mountaineering and climbing, are Olympic disciplines.

The Spanish Government (2022) highlights that, in addition to these being the sports most practiced at the moment, they are also the most egalitarian, since they are practiced almost equally by women and men (49% and 51% respectively), young people and adults from 25 to 54 years of age and over 55 years of age (26% and 32%, respectively) and will have these primary, secondary or university studies (23%, 23% and 32%, respectively), which means note that it is a universal, open and egalitarian sport of mountain sports.

Statistical data from the Higher Sports Council of the Government of Spain show, for the first time in the history of Spanish sport, that hiking and mountaineering are the sports disciplines most practiced by Spaniards. According to the people surveyed, among those who practice sports, a third (30.8%) mainly do hiking and mountaineering, which in terms of the sports population represents 8352402 million Spanish people.

Reviewer 2:

While I can agree that your manuscript is well written, I believe it would greatly profit from some reformatting. Current version is written like a Master thesis. Which is not bad. But for scientific publication, format is bit different. I would like to see more expanded discussion and that you connect with previous and current literature. Now, you just go over your hypothesis in few sentences and that is it. 

Authors:

We appreciate all your comments and suggestions

Discussion and discussion have been expanded (see manuscript), which connects with the literature review and with current studies, including new references to studies from recent years:

Piepiora, P.; BagiÅ„ska, J.; Witkowski, K.; Nakonieczna, J; Piepiora, Z. Comparison of personality differences of Polish mountaineers. Frontiers in psychology 202314, 1120238. https://doi.org/10.3389/fpsyg.2023.1120238

Wulf, G.; Lewthwaite, R. Optimizing performance through intrinsic motivation and attention for learning: The OPTIMAL theory of motor learning. Psychonomic Bulletin & Review 201623(5), 1382-1414. https://doi.org/10.3758/s13423-015-0999-9

Piccoli, A.; Rossettini, G.; Cecchetto, S.; Viceconti, A.; Ristori, D.; Turolla, A.; Maselli, F.; Testa, M. Effect of Attentional Focus Instructions on Motor Learning and Performance of Patients with Central Nervous System and Musculoskeletal Disorders: a Systematic Review. J. Funct. Morphol. Kinesiol. 20183, 40. https://doi.org/10.3390/jfmk3030040

Reviewer 2:

I would put limitation section into discussion and leave Conclusion section purely for your results and implications. Short and concise. Also, mention age in your limitation section. If we are talking about athletes, most of the time we tend not to think of 50 years old. Your results could maybe differ drastically if your mean age was for example 34. 

Authors:

We appreciate all your comments and suggestions.

We have followed their indications, the limitations have been included in the discussion section and we have left the conclusions to deal with the questions they suggest. In relation to age as a limitation, we appreciate your suggestion, we have already dealt with this matter, justifying our position in previous sections.

The practical implications of the study have been included in the final part of the conclusions, following the suggestions of this reviewer and another reviewer. References to studies from recent years have also been included, which support the ideas presented.

Reviewer 2:

Minor comments.

Line 108: Title of the section is: “Effect of emotional intelligence on resilience and satisfaction with life”. Please change satisfaction with life to life satisfaction. 

Authors:

The text has been modified following the suggestion, we appreciate the comment.

Reviewer 2:

Minor comments.

Section 2.1.: typo Parcipants

Authors:

It has been modified following the suggestion, we appreciate the comment.

Reviewer 2:

Table 4.: Missing 1st letter in the first variable.

Authors:

It has been modified following the suggestion, we appreciate the comment.

Reviewer 3 Report

Dear Authors

Thanks a lot for the opportunity you have offered me to revise the fascinating manuscript “Differential magnitude of resilience between emotional intelligence and life satisfaction in mountain sports athletes”.

As a significant strength, this manuscript provides evidence on the potential effect of resilience (personal competence and self- and life-acceptance) on emotional intelligence and life satisfaction in mountain and climbing athletes in Spain. This proposal is a novelty in the field and adds information to the existing evidence in the literature produced in the field.

As a major weakness, the manuscript sometimes needs few details and clarity concerning methodological steps that would help improve the understanding of the manuscript. 

Overall, the paper is well-structured, developed and written. Thus, my peer review is a minor revision. After integrating the improvements, I will be happy to accept it.

#METHODS

*participants: I suggest that the authors provide more details about the participants. Ideally, it would be appropriate to include a table in which the point values (n=...) and frequencies (%) for the different elements assessed such as age, gender, sports would be reported....

*study design: Reporting guidelines were used for this observational study. E.g. STROBE (doi: 10.1371/journal.pmed.0040296). Please explain and possibly organise the reporting accordingly.

#DISCUSSION

*limitations: I suggest moving the section "limitation" from the conclusion to the discussion. Furthermore, I suggest integrating the idea that future studies should investigate the role that certain types of exercise (e.g., with an external focus of attention) may have on improving emotional intelligence and resilience in healthy athletes (doi: 10.3758/s13423-015-0999-9.) and athletes with musculoskeletal pain (doi: 10.3390/jfmk3030040). I suggest the authors consider and integrate these references.

*implications: what are the practical implications for sports people and for future research? I suggest the authors include this part here in the discussions.

Author Response

Reviewer 3 (Review Report, round 2)

Reviewer 3: Black.

Authors: Red.

Reviewer 2:

Thanks a lot for the opportunity you have offered me to revise the fascinating manuscript “Differential magnitude of resilience between emotional intelligence and life satisfaction in mountain sports athletes”.

As a significant strength, this manuscript provides evidence on the potential effect of resilience (personal competence and self- and life-acceptance) on emotional intelligence and life satisfaction in mountain and climbing athletes in Spain. This proposal is a novelty in the field and adds information to the existing evidence in the literature produced in the field.

As a major weakness, the manuscript sometimes needs few details and clarity concerning methodological steps that would help improve the understanding of the manuscript. 

Overall, the paper is well-structured, developed and written. Thus, my peer review is a minor revision. After integrating the improvements, I will be happy to accept it.

Authors:

We appreciate all your comments and suggestions.

Reviewer 2:

Method

Participants: I suggest that the authors provide more details about the participants. Ideally, it would be appropriate to include a table in which the point values (n=...) and frequencies (%) for the different elements assessed such as age, gender, sports would be reported, ...

Authors:

We appreciate all your comments and suggestions.

Between lines 155 to 170, the characteristics and sociodemographic details of the sample are presented:

“… In relation to gender distribution, 2696 subjects were male (67.1%) and 1322 were female (32.9%). The mean age of the participants was 49.42 years old (±11.9).

In relation to the type of discipline they practice within mountain sports, 31.68% practice hiking, mountaineering 15%, mountain running 11.36%, climbing or para-climbing 10.03%, climbing in climbing walls 7. 59%, snowshoeing 5.32%, ski mountaineering 7.25%, canyoning 3.35%, Nordic walking 2.12%, ice climbing 1.52%, snow running 0.49%, and 4.02% practice other mountain disciplines”.

It has been considered that it was more appropriate to report the characteristics of the sample, in this way to avoid overloading the article with tables (following the recommendations of the Academic Editor, part of the tables have been included in the "Supplementary Material").

Reviewer 2:

Study design: Reporting guidelines were used for this observational study. E.g. STROBE (doi: 10.1371/journal.pmed.0040296). Please explain and possibly organise the reporting accordingly.

Authors:

We appreciate all your comments and suggestions. These suggestions have been taken into account.

Reviewer 2:

Discussion

Limitations: I suggest moving the section "limitation" from the conclusion to the discussion. Furthermore, I suggest integrating the idea that future studies should investigate the role that certain types of exercise (e.g., with an external focus of attention) may have on improving emotional intelligence and resilience in healthy athletes (doi: 10.3758/s13423-015-0999-9.) and athletes with musculoskeletal pain (doi: 10.3390/jfmk3030040). I suggest the authors consider and integrate these references.

Authors:

The limitations have been included in the "Discussion" section (lines 330-341). The idea that future studies should investigate the role that certain types of exercise may have in improving emotional intelligence and resilience in healthy athletes has been integrated, including relevant references [75-76]:

Wulf, G., & Lewthwaite, R. Optimizing performance through intrinsic motivation and attention for learning: The OPTIMAL theory of motor learning. Psychonomic bulletin & review 201623(5), 1382-1414. https://doi.org/10.3758/s13423-015-0999-9

Mention has also been made of its influence with musculoskeletal aspects:

Piccoli, A.; Rossettini, G.; Cecchetto, S.; Viceconti, A.; Ristori, D.; Turolla, A.; Maselli, F.; Testa, M. Effect of Attentional Focus Instructions on Motor Learning and Performance of Patients with Central Nervous System and Musculoskeletal Disorders: a Systematic Review. J. Funct. Morphol. Kinesiol. 20183, 40. https://doi.org/10.3390/jfmk3030040

Reviewer 2:

Implications: what are the practical implications for sports people and for future research? I suggest the authors include this part here in the discussions.

Authors:

We appreciate all your comments and suggestions.

Practical suggestions from this research demonstrated that improving athletes' emotional intelligence could have a positive impact on their psychological well-being and life satisfaction. In this sense, specific emotional training programs could be developed for athletes that address skills such as recognition and regulation of emotions, empathy and assertiveness. Similarly, the importance of resilience as a key ability to face and overcome challenges and adversities in the context of mountain and climbing sports. Wulf and Lewthwaite [75]. Resilience training strategies could be implemented to help athletes develop greater adaptive and recovery capacity, following Piccoli et al. [76].

Other reviewers have suggested that limitations be included in the conclusions section at the end of the article, which is why they are included at the end of the study.

Conclusions are improved and expanded including:

The findings of this research may be relevant in terms of psychological evaluation in the sports field, not only focused on physical performance, but also on emotional and well-being aspects. Valid and reliable measurement instruments could be used to assess the emotional intelligence, resilience and life satisfaction of athletes, which would make it possible to identify areas for improvement and personalized design, until life satisfaction is a key aspect for the general well-being of athletes. Coaches and sports managers could consider strategies that foster a positive environment and motivate in the sports context, in addition to providing emotional support to athletes to improve their satisfaction with sports practice.
